# Aplastic Internal Carotid Artery: A Potentially Catastrophic Vascular Anomaly

**DOI:** 10.3390/diagnostics13193089

**Published:** 2023-09-29

**Authors:** Benjamin L. Bosse, Geoffrey Wilkinson, Zoe N. Anderson, Jay Babu, Riyaa Rajesh, Rajesh Rangaswamy, Karthikram Raghuram

**Affiliations:** 1School of Medicine, University of Nevada, Reno, NV 89557, USA; benbosse@med.unr.edu (B.L.B.); gwilkinson@med.unr.edu (G.W.); 2Kirk Kerkorian School of Medicine, University of Nevada, Las Vegas, NV 89106, USA; babuj1@unlv.nevada.edu; 3Reno Radiological Associates, Reno, NV 89434, USA; rajeshriyaa@gmail.com (R.R.); rrangaswamy@renorad.com (R.R.); kraghuram@renorad.com (K.R.)

**Keywords:** ICA, mural thrombus, MRI, CTA, neuroradiology

## Abstract

Congenital absence of an internal carotid artery (ICA) is a rare vascular anomaly and occurs in less than 0.01% of the population. We report a case of aplastic internal carotid artery in a 34-year-old female. The patient presented to the emergency department with complaints of new-onset involuntary swaying-like movement of her right arm. Brain magnetic resonance imaging showed multifocal tiny areas of acute infarcts in the bilateral frontal, parietal, and left occipital lobes in the watershed distribution. There was no visualization of the flow of the intracranial left internal carotid artery. Follow-up CTA of the head and neck showed a congenital absence of the left internal carotid artery with no evidence of arterial dissection, occlusion, or aneurysm. Obstruction of the internal carotid artery has significant consequences for patients. This effect is amplified if the disruption occurs in the sole anterior blood supply to the parenchyma of the brain, as in this case. In our patient care, imaging was vital to the detection and subsequent treatment with anticoagulation to avoid further cerebral complications, and the patient will now have a better understanding of the increased lifetime risk of further events.

A 34-year-old female with a past medical history of GERD presented to the emergency department with complaints of new-onset involuntary swaying-like movement of her right arm that began 30 min prior to her presentation. The patient stated that she had COVID-19 six months prior without any cough or congestion, just a fever. Over the three months prior to presentation, she had a persistent dry cough, dyspnea on exertion, and orthopnea. In the two weeks prior to presentation, she noted bilateral lower extremity edema up to the upper thighs. She also was experiencing associated shortness of breath, headache, and fatigue.

On physical exam, the patient was an obese woman in no acute distress. She was normocephalic and atraumatic with moist mucous membranes. Extraocular movements were intact, and her pupils were equal, round, and reactive to light. Her cardiac exam demonstrated distant heart sounds with a tachycardic rate and regular rhythm. Her lungs were significant for decreased breath sounds bilaterally and normal pulmonary effort. Her abdomen was soft and non-tender to palpation. The patient had noted 2+ pitting edema in her bilateral lower extremities. Her neurologic exam showed no focal deficits, and the patient was oriented to person, place, time, and situation.

**Figure 1 diagnostics-13-03089-f001:**
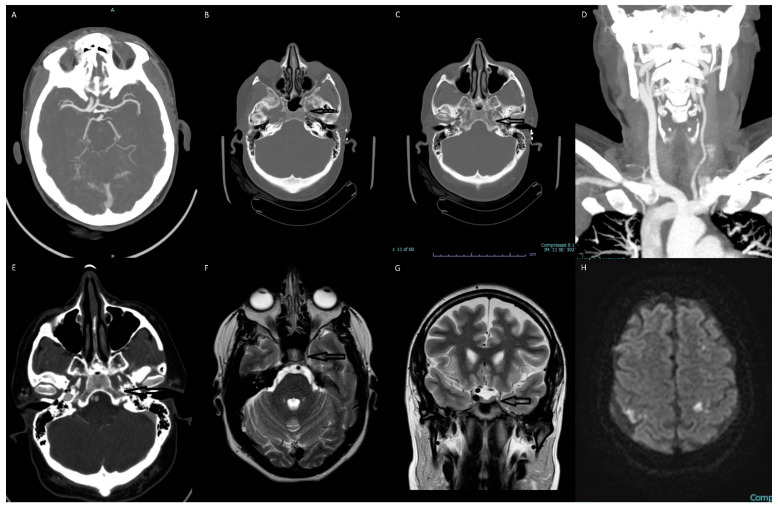
Multiple images depict the absence of the left internal carotid artery (ICA). (**A**) The transverse section of the CTA of the brain shows an absent supraclinoid segment of the ICA on the left side. The left anterior and middle cerebral arteries are reconstituted via the anterior communicating and posterior communicating arteries. (**B**) The axial section of the brain CT scan shows an absent horizontal petrous segment of ICA on the left side. (**C**) The axial section of the brain CT scan shows an absent vertical petrous segment of ICA on the left side. (**D**) Coronal MIP CTA showing arterial branching from the aortic arch and an absent left ICA. (**E**) The axial section of the CTA of the brain shows an absent vertical petrous segment of ICA on the left side. (**F**) The axial T2 weighted MR image shows absent flow void of left ICA. (**G**) The coronal T2 weighted MR image shows absent flow void of left ICA. (**H**) The brain MR diffusion-weighted imaging (DWI) showing focal regions of ischemia in bilateral cerebral hemispheres.

The most significant findings of the patient’s laboratory workup were an elevated troponin of 336 and a proBNP of 19,572. Her laboratory studies also demonstrated mild hypokalemia at 3.3, elevated BUN at 29, a creatinine of 1.47, a mildly elevated ALT of 53, and an INR of 1.18. Imaging studies included: Dx-Chest, Echocardiogram, MR-Brain-W/O, MRV Head, CT-CTA Neck with and W/O, CT-CTA head with and W/O. The patient’s echo showed severely reduced left ventricular systolic function with an ejection fraction of 30%, apical akinesis, and a thrombus present in the left ventricular apex. Brain magnetic resonance imaging showed tiny multifocal areas of acute infarcts in the bilateral frontal, parietal, and left occipital lobes in the watershed distribution. There was no visualization of the flow of the intracranial left internal carotid artery. The follow-up CTA of the head and neck showed a congenital absence of the left internal carotid artery with no evidence of arterial dissection, occlusion, or aneurysm.

The patient’s neurologic deficits with the presence of seizures were linked to the presence of a thrombus in the left ventricle that had shower embolized. Given the distribution of strokes in both cerebral hemispheres, the mechanism of the stroke was most likely embolic, from the apical cardiac thrombus. She was started on a heparin infusion and subsequently transitioned to Lovenox, while simultaneously being started on warfarin until a therapeutic range of INR was obtained. The target therapeutic INR was 2–3. The patient was discharged two days later with therapeutic warfarin and 750 Keppra BID and was lost to follow-up.

**Figure 2 diagnostics-13-03089-f002:**
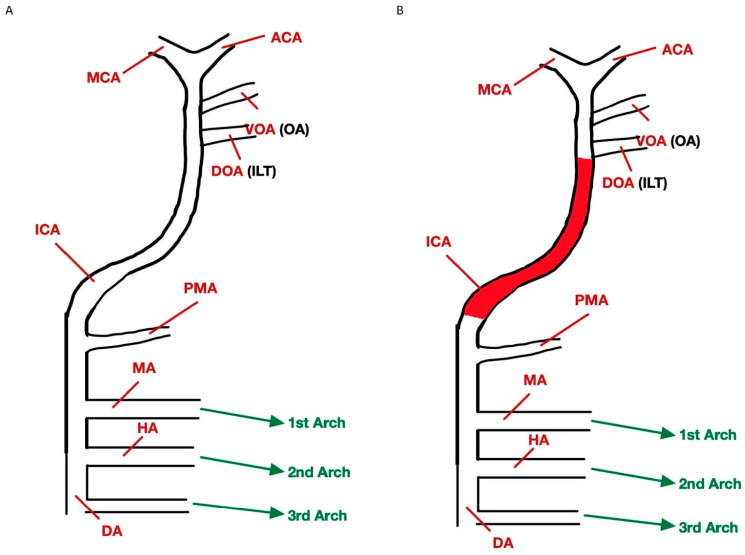
Depiction of normal embryologic arterial anatomy (**A**) and embryologic arterial anatomy that failed to form (**B**). Internal carotid artery (ICA), middle cerebral artery (MCA), anterior cerebral artery (ACA), dorsal aortic arch (DA), hyoid arch (HA), mandibular arch (MA), primitive maxillary artery (PMA), dorsal ophthalmic artery (DOA), inferolateral trunk (ILT), ventral ophthalmic artery (VOA), and ophthalmic artery (OA).

Congenital absence of an internal carotid artery (ICA) is a rare vascular anomaly and occurs in less than 0.01% of the population [1]. Agenesis, aplasia, and hypoplasia are other reported ICA abnormalities [1]. In congenital absence of ICA, the collateral circulation that does develop is dependent on the embryonic period in which cessation of the ICA occurred [1]. In our case, the patient had collateral flow via the circle of Willis, which forms during the 7–24 mm stage of embryonic development [1]. During early embryonic development, the internal carotid artery is the sole source of blood supply for the developing brain, until the posterior circulation develops later [2]. When there is an absence of an ICA, collateral circulation via the circle of Willis, persistent embryonic arteries, or transcranial collaterals of the external carotid artery develop [1]. An exact cause for these abnormalities has yet to be identified, although they are thought to represent consequences of an insult to the developing embryo [3]. Potential insults include mechanical or hemodynamic causes in early development, such as amniotic adhesions, mechanical causes, pressure effects, or excessive bending of the embryo [3]. Given the adaptive process of increased embryological collateral circuit production, many cases are diagnosed incidentally [1]. The manifestations of aplastic ICA consist of cerebrovascular events including acute ischemia, subarachnoid hemorrhage, cerebral aneurysm, and parenchymal hemorrhages [1]. Neurologic deficits such as cranial nerve deficits, vision disturbances, hemiparesis, dysphagia, seizures, pulsatile tinnitus, migraine, and Horner’s syndrome have also been reported [4,5]. Obstruction of the internal carotid artery has significant consequences for patients. This effect is amplified if that obstruction occurs in the sole blood supply to the parenchyma of the brain. In the event that atherosclerotic disease affects the remaining normal carotid artery, both cerebral hemispheres would be at risk for infarction.

**Figure 3 diagnostics-13-03089-f003:**
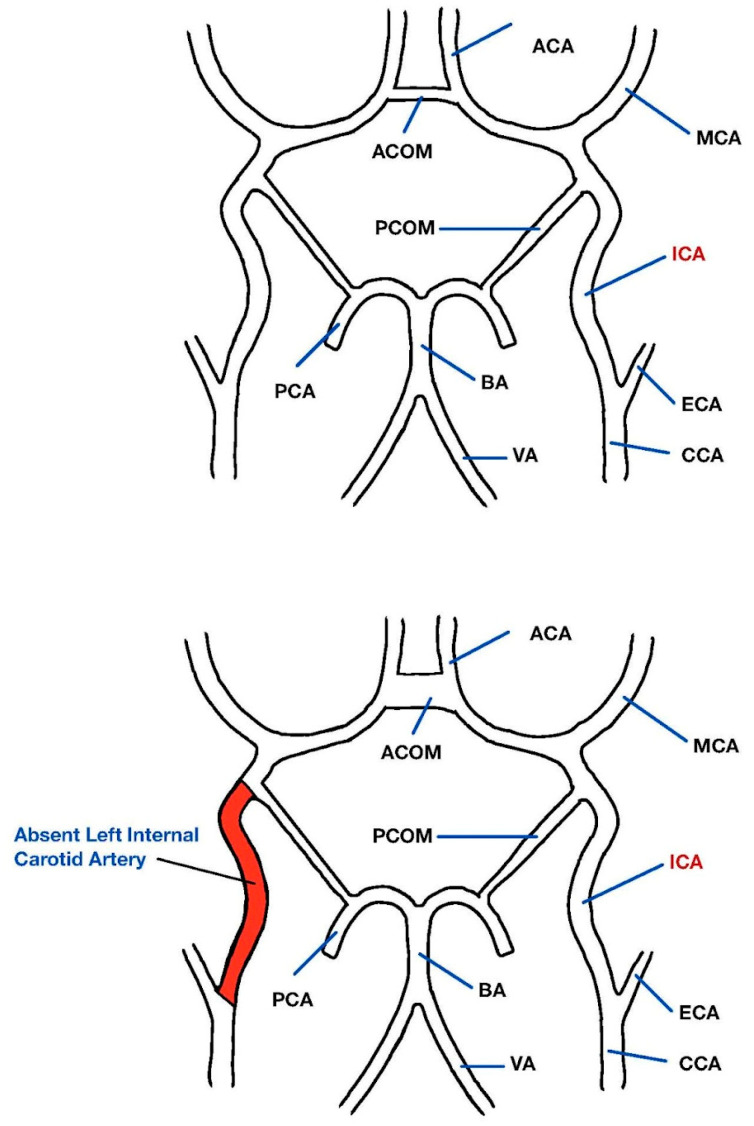
Top graphic demonstrating normal anatomy of the circle of Willis and surrounding vasculature. Bottom graphic demonstrating absent left internal carotid artery with no blood flow through the highlighted red segment. Internal carotid artery (ICA), middle cerebral artery (MCA), anterior cerebral artery (ACA), anterior communicating artery (ACOM), posterior communicating artery (PCOM), posterior cerebral artery (PCA), basilar artery (BA), vertebral artery (VA), external carotid artery (ECA), common carotid artery (CCA).

## Data Availability

Not applicable.

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
