# Peer review of "Aplastic Internal Carotid Artery: A Potentially Catastrophic Vascular Anomaly"

_diagnostics, 2023, doi:10.3390/diagnostics13193089_

Round 1

Reviewer 1 Report

Please provide a CTA  reformatted image for the aortic arch and extracranial vessels to visualise the full extent of the anomaly.

It would also be of value to provide a TOF MRA image of the intracranial vasculature.

Otherwise this is an interesting case.

Adequate

Author Response

1) New images added to paper.  

Reviewer 2 Report

Thanks for the opportunity to review this work. Here are a few comments:

-          Please define « carotid disruption » and the mechanism that caused the patient to have multiple infarcts

-          DWI + ADC images of the infracts would be interesting to add

-          Images depicting the disease found in the right carotid artery (the disruption) that caused stroke should be added and commented

-          There seems to be discrepancy between what the abstract tells us (“disruption of the carotid”), and the main text (apical thrombus in the LV that caused multiple emboli), please check and revise the structure of both the abstract and main text

-          Any data on the patient’s long-term outcome?

-          Please briefly discuss treatment options and clinical implications (screening for intracranial aneurysms?)

-          The anatomical and embryological discussion is helpful and well presented!

Author Response

1) Deleted Carotid Disruption and changed to aplasia of ICA. 

2) Elaborated on embolic nature.

3) Added update on follow-up

4) Mentioned clinical implications

Reviewer 3 Report

Development of the internal carotid artery is not from single pipe, it consists of several segment. Before conclusion of which segment is missing, vascular imaging should be demonstrated more, at least external carotid artery should be identified.  Cerebral angiogram is the best way, otherwise, good quality of CTA or MRA can be alternative.

Author Response

1) Cerebral angiogram was not performed, as the diagnosis was made with a high degree of confidence on the CTA and MRA.

Reviewer 4 Report

The publication does not follow the typical scientific format with an “introduction”, “case description”, and “discussion”.

Figures 2 and 3 appear to be low-quality and unprofessional. To improve their appearance, try using a free image editor like "Photopea" or "GIMP" if you don't have access to Photoshop.

The publication's scientific value could be enhanced by including additional references, despite the already well-chosen ones.

Improving the quality of the language used in the publication will undoubtedly enhance the overall quality and readability of the article.

The attached file includes all of the necessary corrections.

Author Response

1) Interesting images case, so we were prompted to utilize a different format

2) Thank you for the grammar suggestions!

Round 2

Reviewer 3 Report

I suggested to show the images of neck arteries to visualize the CCA and ECA and to confirm agenetic or hypoplastic ICA 

Usually, this anomaly is asymptomatic in adult, so the neurological presentation in this patient may not result from the anomaly

The discussion should be included the embryology of the ICA, so the readers can understand how ICA could be total agenesis or segmental agenesis and the common associated anomaly which can be found.

Author Response

Added to our discussion of the embryology. 

Included an image of neck vessels

The thrombus was of cardiac origin. Thus, future risk would likely be from atherosclerosis, not the cardiac thrombus. 

Round 3

Reviewer 3 Report

1. It's good to have the coronal picture to seen patency of the left CCA and ECA, however, I think it should be better if you could add the sagittal plane to see at the CCA bifurcation to demonstrate that no ICA and the CCA is continue to be ECA

2. I don't agree with your discussion in the sentences 101-106, most of the lesion seen in adult is asymtomatic because the colaateral circulation will be develop and adapt quite well, if it will cause symptom of inadequate flow it should present since childhood.

Author Response

Correction #1 Thank you for the comment, and we appreciate that more imaging would always be preferable. We feel that the coronal image aptly displays branches of the left ECA while highlighting the lack of left ICA. The imaging that we included was focused on an absence of the left ICA and we feel that it would be better to continue to focus on that aspect of the case.   Correction #2 We appreciate your comment regarding the validity of the statement we made and agree that most of these anatomic variants are asymptomatic. However, we suggest that in the event that there is an atherosclerotic disease or another occlusive issue, this would have a much more profound effect in this patient. We believe the wording of this section accurately reflects this position.